# Dysregulated Expression Patterns of Circular RNAs in Cancer: Uncovering Molecular Mechanisms and Biomarker Potential

**DOI:** 10.3390/biom14040384

**Published:** 2024-03-22

**Authors:** Nicole R. DeSouza, Kate J. Nielsen, Tara Jarboe, Michelle Carnazza, Danielle Quaranto, Kaci Kopec, Robert Suriano, Humayun K. Islam, Raj K. Tiwari, Jan Geliebter

**Affiliations:** 1Department of Pathology, Microbiology and Immunology, New York Medical College, Valhalla, NY 10595, USA; ndesouza@student.nymc.edu (N.R.D.);; 2Division of Natural Sciences, University of Mount Saint Vincent, Bronx, NY 10471, USA; 3Department of Otolaryngology, New York Medical College, Valhalla, NY 10595, USA

**Keywords:** circular RNAs, circRNA, circRNA function, circRNA application, cancer

## Abstract

Circular RNAs (circRNAs) are stable, enclosed, non-coding RNA molecules with dynamic regulatory propensity. Their biogenesis involves a back-splicing process, forming a highly stable and operational RNA molecule. Dysregulated circRNA expression can drive carcinogenic and tumorigenic transformation through the orchestration of epigenetic modifications via extensive RNA and protein-binding domains. These multi-ranged functional capabilities have unveiled extensive identification of previously unknown molecular and cellular patterns of cancer cells. Reliable circRNA expression patterns can aid in early disease detection and provide criteria for genome-specific personalized medicine. Studies described in this review have revealed the novelty of circRNAs and their biological ss as prognostic and diagnostic biomarkers.

## 1. Introduction

Circular RNA (circRNA) molecules represent a unique class of non-coding RNAs (ncRNAs) that exert their biological roles as stable, covalently closed, RNA transcripts that orchestrate a multitude of cellular functions. NcRNAs are classified based on structure and origin of biogenesis, which significantly determines their functional mechanisms. CircRNAs are classified into four main categories, including exonic, intronic, exon–intron, and intergenic [1]. Unlike linear ncRNAs, circRNAs represent pre-mRNAs that undergo a back-splicing process, forming an RNA molecule with joined 3′ and 5′ ends [2]. Due to their enclosed structure, circRNAs are resistant to RNases, enabling long half-lives and reliable expression patterns in various fluid mediums (i.e., blood, urine, gastric juices, lymph, etc.) [3]. This premise alone makes circRNAs attractive biomarkers for disease. In the last decade, the data have supported a significant role for ncRNAs in the development of many pathological conditions, especially cancer [4,5,6,7]. The up- and downregulation of ncRNAs have been identified as potent drivers of dysregulated protein-coding gene expression. This is attributed to their extensive, multi-ranged ability to orchestrate epigenetic modifications, drastically impacting carcinogenic phenotypes [8]. CircRNAs work as cis genome regulators, facilitated by their structural capacity to bind complementary nucleic acid sequences, as well as a single protein or multiple proteins, simultaneously [9]. For example, circRNAs serve as decoys for microRNAs (miRNAs) and RNA-binding proteins, resulting in the modulation of both gene expression and post-translational functions [10] (Figure 1). Protein-binding interactions are classified as “protein-centric” or “circRNA-centric”, which refer to circRNAs bound by a specific protein, or the binding of proteins to a specific circRNA, respectively [11]. Due to these dynamic functions, it has been reported that certain circRNAs can confer drug resistance, or aid in the acquisition of therapeutic resistance mechanisms in cancer cells [12]. Thus, dysregulated expression of circRNAs that exert these full-ranged cellular functions enabled their emergence as promising biomarkers for early detection of cancer, patient prognosis, and therapeutic regimen. 

Recent research has focused on circRNA targeting as a putative therapeutic approach [13,14]. Genome-wide studies of a multitude of cancer types have revealed active pathologic roles for circRNA molecule expression and genomic modulation. It has been described in various studies in the literature that circRNAs are actively contributing to carcinogenic propagation and progression [15,16]. ROC analysis recommendations are made in the literature to aid in determination of a circRNA’s potential as a putative biomarker [17]. Particularly, their function as molecular sponges has placed them in a central role for fine-tuning genomic output—greatly impacting genome-wide instability employed by cancer cells to promote their survival (Figure 2). 

Although rapidly progressing, additional research is warranted for the investigation of circRNAs as emerging novel targets. The identification of cancer-type-specific molecular mechanisms by which circRNAs promote carcinogenesis can uncover actionable therapeutic targets, as well as prognostic and diagnostic markers [18]. Bioinformatic analysis of transcriptomic datasets in combination with predictive software for RNA interaction have extensively enabled the identification of transcriptomic targets for evaluation (Table 1). 

Understanding differential gene expression patterns that arise in cancerous vs. non-cancerous tissue, different tumor stages of the same malignancy, as well as amongst treatment-responsive patients within cancer types, has significantly contributed to the ingenuity of cancer genomics (Figure 3). Additionally, use of gene editing technologies implemented in vitro and in vivo has further enabled evaluation of putative markers. This review will focus on the role of circRNAs of selected representative cancers and their identification process through bioinformatics, as well as their role in the onset and progression of carcinogenesis. 

## 2. CircRNAs: Methods for Study and Evaluation

As mentioned, advances in computational methods have significantly expanded in silico gene analysis. Sequence complementarity and functional evaluation of nucleic acid regions have enabled the ability for successful predictions of identified circRNA sequences (Table 1). The adaptation of in vitro methodologies for functional circRNA analysis has enabled the identification of novel cell mechanisms as well as novel cell targets. As stated, circRNAs have a characteristic structure that distinctly separates them from other RNA molecule counterparts. It is imperative to establish that the identified molecule is a bona fide circRNA—eliminating overlap with similar mRNAs, linear non-coding RNAs, and pseudo-splice variants. These molecules, therefore, utilize their RNA-and protein-binding domains to orchestrate cellular function. Methods for identification of these processes have significantly expanded and contributed immensely to further evaluation of epigenetic and proteomic regulation in different pathologies. 

### 2.1. Identification of circRNA–Protein Interactions

Due to the enclosed structure of circRNAs, there are unique interactions with RNA-binding proteins (RBPs) that differ from protein-bound linear RNA molecules. These dynamic interactions rely on the tertiary circular structure of circRNAs, and are extremely versatile in influence—i.e., some interactions regulate circRNA expression, whereas others regulate bound protein expression. Thus, the fate of these interactions heavily relies on tissue specificity as well as the putative functions of the unbound substrates [29,30,31].

It has been reported that circRNA–protein interactions have significantly contributed to cellular mechanisms, namely, proliferation, apoptotic induction, invasion, migratory propensity, angiogenesis, transcription, translation, and differentiation [32]. Stated above, circRNAs interact with proteins in either a protein-centric or self-centric mechanism. Protein-centric methods are employed to identify bound proteins to circRNAs. These methods are used when a protein of interest is known and circRNA antibody-binding domains occupied by a protein of interest can be evaluated. RNA immunoprecipitation (RIP) procedures are employed to evaluate circRNA regions predicted to bind proteins. These regions are evaluated with RT-qPCR with primers generated for known circ-junctions, or the identification of circ-junctions is carried out using bioinformatic analysis, predictive algorithms, or RNA sequencing. Further exploration of circ-junctions can be accomplished with Electrophoretic Mobility Shift Assay (EMSA). This process involves the comparison of bound and unbound circRNA molecules to identify if protein interactions are present. CircRNA molecules bound can be further identified following protein-centric methodology using sequencing analysis [33]. 

CircRNA-centric methodology would involve the use of RNA pull-down assays, which involves the identification of proteins bound to a known circRNA. This is a commonly used method for identifying proteins bound to a linear RNA molecule; however, treatment of a sample with RNase R enables exclusive evaluation of bound proteins to circRNAs. Following pull-down, bound proteins can be identified with mass spectrometry, or quantitatively compared using Western blot once identified. The latter identification method is commonly used when studies over- or under-express circRNAs for functional evaluation. Secondary evaluation of proteins identified with mass spectrometry with a comparative Western blot analysis has become common practice to avoid non-specific binding partners of circRNAs during pull-down [34].

For general visualization of known circRNA and protein interactive partners, once identified, fluorescence in situ hybridization (FISH) can be implemented to label the circRNA molecule, followed by immunofluorescence procedures to visualize a target protein. Taken together, both labelling processes can visualize circRNA–protein physical interaction [31]. 

### 2.2. Identification of circRNA–miRNA Interactions

RNA–RNA interactions can be predicted based on sequence complementarity and hybridization energy. Many algorithms have therefore been used for software production that can enable accurately predicted RNA interactors for further in vitro assessment [35,36] (Table 1). Gene expression analysis is a common method used to explore predicted miRNA targets of circRNAs. The over- or underexpression of circRNAs will lead to the modulation of target miRNA expression. To identify if this interaction is directly related, FISH can be used to individually tag the circRNA and miRNA of interest and the association can be visualized. As mentioned, regarding protein-centric methodology, RNA pull-down can similarly be used to identify bound miRNAs, which can then be identified through RNA sequencing, if unknown, or qRT-PCR, for comparative expression evaluation when circRNAs are over- or underexpressed [37]. 

## 3. Analysis of circRNA Functions in Cancer Acute Myeloid Leukemia

Acute myeloid leukemia (AML) is a cancer of the blood and bone marrow. Common in adults, AML originates from the disordered proliferation of immature myeloid progenitors followed by blockage of differentiation. AML is an extremely genetically diverse and aggressive hematological malignancy [38]. Although there have been therapeutic advancements and significant rates of remission, greater than 50% of AML patients experience short-term relapse and reduced responsiveness to treatment. Studies have shown that the FMS-like tyrosine kinase 3 (FLT3) gene is often overexpressed in hematopoietic neoplasms, which is frequently mutated in AML patients. The FLT3-ITD mutation is caused by an internal tandem duplication (ITD) of the juxtamembrane domain-coding region, inducing auto-phosphorylation at tyrosine residue Y591, driving constitutive activation of related signaling pathways. The presence of the FLT3-ITD mutation correlates with a very poor prognosis and higher rate of relapse after a shorter period of remission when compared to other AML subtypes. FLT3 inhibitors have demonstrated single agent efficacy; however, acquired resistance with secondary tyrosine kinase domain (TKD) mutations illustrates the need for additional therapeutic targets [39]. Further, circRNA molecule expression is extensively understudied in leukemia-specific disease, warranting the identification of novel circRNAs as well as the identification of novel roles in leukemia for known circRNAs found in other cancer types. 

### 3.1. Role of circMYBL2 in Acute Myeloid Leukemia

The circRNA circMYBL2 (also known as hsa_circ_0006332) regulates FLT3 translation by recruiting PTBP1 (polypyrimidine tract binding protein 1), a process that can promote the FLT3-ITD mutation found in AML. Thus, circMYBL2 plays a key role in the progression of FLT3-dependent leukemia. Knockdown verification of this circRNA considerably impaired the activity within FLT3-ITD AML cells, including those resistant to FLT3 inhibitors. A published study evaluated 28 AML patients and their circMYBL2 levels. Patients with FLT3-ITD^+^ and FLT3-ITD^−^ AML showed significant differences in circMYBL2 levels; FLT3-ITD^+^ patients had higher levels of circMYBL2 expression. Further research evaluated six FLT3-ITD^+^ AML patients’ samples and 45 FLT3-ITD^−^ AML patients’ samples. Approximately five-fold higher expression levels of circMYBL2 in FLT3-ITD^+^ AML patient samples were observed than in the FLT3-ITD^−^ AML patient samples. Collected data indicates that the identification of circMYBL2 partnered with FLT3-ITD^+^ AML patients has the potential to contribute to leukemia progression. In summary, this research provides the prospect of circMYBL2 as a possible therapeutic target for this AML subtype presenting with the FLT3-ITD mutation [39]. 

### 3.2. Role of hsa_circ_0075451 in Acute Myeloid Leukemia

A cohort study by Wang et al. analyzed circRNA transcriptomic data from 60 cytogenetically normal AML (CN-AML) patients for the identification of novel diagnostic predictors. This study identified 308 circRNA candidates that served as predictors of overall survival. Specifically, hsa_circ_0075451 expression was found to serve as an independent predictor, with a further identified regulatory network of 84 hub genes it potentially regulates. As stated, molecular markers have been well established and identified in AML (i.e., mutations such as *DNMT3A*, *IDH1*, and *IDH2*); therefore, they were manipulated in order to evaluate if hsa_circ_0075451 had any contributing role to their expression or impact on disease development and progression. Wang et al. assessed the mRNA expression profiles of 14 CN-AML patients, half of whom had high hsa_circ_0075451 expression, and the remaining half had low hsa_circ_0075451 expression. Between these two groups, 775 differentially expressed genes were found, many of which are involved in the regulation of Wnt signaling pathways. Additional in silico analysis was used to identify a circRNA/miRNA/mRNA network. Wang et al. reported that hsa_circ_0075451 was predicted to bind several miRNAs, such as miR-515-5p, miR-873, miR-766-3p, miR-940, miR-661, miR-492, miR-330-5p, miR-326, miR-512-5p, and miR-338-3p. At the interface of regulation, mRNA transcript PRDM16 was found to be regulated by eight of these miRNAs (miR-873, miR-940, miR-766-3p, miR-661, miR-492, miR-330-5p, miR-338-3p, and miR-326), and was further found to be positively correlated with hsa_circ_0075451 expression in the CN-AML patient cohort. They also reported that hsa_circ_0075451 expression was negatively correlated with the identified miRNAs, warranting in vitro assessment of expression function. Luciferase reporter assays were used, and decreased luciferase activity was found in cells with two experimentally established phenotypes: cells co-transfected with candidate miRNA mimics (miR-326, miR-330-5p, and miR-338-3p), and also with either wild type or mutant hsa_circ_0075451. These miRNA candidates were found to be bound by hsa_circ_0075451, which was further validated by a shared regulatory interaction with PRDM16. RNA-FISH confirmed a direct co-interaction and localization of hsa_circ_0075451 with miR-330-5p, demonstrating that hsa_circ_0075451 and miR330-5p together regulate the PRDM16 locus. This study served as a critical first step in the identification of the ability for hsa_circ_0075451 to serve as an extensive molecular sponge for a plethora of miRNAs that tightly regulate the transcription of oncogenes, driving AML progression [40]. 

### 3.3. Role of hsa_circ_004277 in Acute Myeloid Leukemia

Li et al. performed a circRNA microarray to establish the expression profiles in a cohort of six newly diagnosed (i.e., not yet treated) AML patient samples compared to healthy controls. This study evaluated the function of the most significantly downregulated circRNA, hsa_circ_004277, which was found to negatively promote AML in a leukemia-specific manner. Further expression evaluation of 107 AML patients of varying disease stages identified a lower expression pattern of hsa_circ_004277 in patients that had not yet received any form of therapy. Additionally, expression patterns of hsa_circ_004277 were shown to be higher in patients that had achieved complete remission, potentially elucidating an upregulation of hsa_circ_004277 following successful chemotherapy. In accordance with this notion, hsa_circ_004277 expression levels were downregulated in patients with cancers that had relapsed/were refractory to chemotherapy. This observation provided a significant example of the dynamic expression patterns of circRNAs within a cancer type, providing evidence for the massive potential these molecules possess as diagnostic and/or prognostic markers. With this understanding that fluctuating expression levels correlate to patient responses and outcomes, identification of the mechanisms causing this apparent relationship between the two is critical and could suggest additional genomic targets [41]. 

## 4. Multiple Myeloma

Multiple Myeloma (MM) is a clonal plasma cell malignant neoplasm and is the second most common hematological malignancy. MM originates from the post-germinal lymphoid B-cell lineage. The disease typically evolves from the anomalous proliferation of clonal plasma cells, termed “monoclonal gammopathy of undetermined significance” (MGUS) [42]. MM has disproportionate incidence in the African American population, with incidence twice that of the European American population and is predominantly diagnosed in those over 60 years of age. Frequently, MM is correlated with notable morbidity because of its end-organ destruction [43]. Yearly, an estimated ~35,000 MM cases will be diagnosed in the US, with a slightly higher risk rate in males. In the past decade, diagnosis and therapeutic advancements have led to improved overall survival and revisions regarding the staging system for MM. Treatment for MM includes traditional therapies, namely chemotherapies, radiation therapy, and bone marrow transplantation. Additionally, novel immunotherapies are being utilized to treat MM, notably chimeric antigen receptor T cell (CAR-T cell) therapy, in which autologous cells are taken from the host body and modified to bind to cancerous cells and kill them [44]. The necessity for further therapeutic intervention is clear in MM. As such, research has recently focused on the investigation of circRNAs closely related to incidence and progression of malignant hemopathies [45]. 

### 4.1. Role of circCHEK1_246aa in Multiple Myeloma

Chunyan et al. demonstrated the promising therapeutic usage of circCHEK1_246aa as target for both MM and bone marrow (BM) niches. The BM microenvironment has been shown to support MM cell survival and drug resistance. Checkpoint Kinase 1 (CHEK1) was chosen for evaluation because of its promising therapeutic potential [46]. This is based on previous studies that show its inhibition can suppresses KRAS mutation-driven cancers. Previous pharmacological research established the efficacy of CHEK1 inhibitors for the treatment of MM. Moreover, CHEK1 plays a vital role in other cancers, not only MM, including lung and uterine cancers [47,48]. The newly discovered circCHEK1_246aa was found to potentially to promote MM proliferation and osteoclast differentiation when secreted into the BM microenvironment. Additionally, Chunyan identified downstream targets of CHEK1, supplying noticeable insight into CHEK-dependent mechanisms of MM malignancy and bone lesion formation. Assessment of circCHEK1_246aa existence was completed through qPCR, Sanger sequencing, and mass spectrometry. Results concluded transfection of circCHEK1_246aa increased MM chromosomal instability (CIN) and osteoclast differentiation. Correspondingly, CHEK1 overexpression confirmed that MM cells secrete circCHEK1_246aa in the BM niche to increase the invasive potential of MM cells and promote osteoclast differentiation. Overall, circCHEK1_246aa has propitious therapeutic potential for MM.

### 4.2. Role of circ_0000190 in Multiple Myeloma

A study performed by Feng et al. identified the location of circ_0000190 and its role in regulating the progression of MM. Modern studies show that dysregulation of miRNAs and long non-coding RNAs have significant impacts on progression of MM. Microscopic examination through fluorescent in situ hybridization was used to successfully locate circ_0000190 within the MM cell [49]. Furthermore, qRT-PCR and Western blot were both utilized to evaluate protein and RNA expression. In vitro cell viability, proliferation, apoptosis assays and flow cytometry were conducted to assess the effects of circ_0000190 and its miRNA target on MM. MiRNA target, miR-767-5p, was selected through computational screening and a luciferase reporter assay. Luciferase reporter assay confirmed the binding ability between circ_0000190 and miR-767-5p. A mouse model of human MM was established using subcutaneous xenograft tumors. Circ_0000190 testing revealed its cytoplasmic localization, as well as its downregulation in BM tissue and peripheral blood. Feng et al. concluded that circ_0000190 inhibited the multiple central carcinogenic mechanisms of MM, including cell viability and proliferation, thus suppressing cell progression. Circ_0000190′s successful inhibition is possible through negative regulation of miR-767-5p within the mitogen-activated protein kinase 4 (MAPK4) pathway. MAPK4 is a direct target of miR-767-5p; by targeting and regulating the MAPK4 pathway, miR-767-5p is capable and proficient in promoting tumor growth as well as progression. Circ_0000190 retains the ability to aid in protection against MM, through repression of miRNA target mi-767-5p. Conclusively, circ_0000190 might possess strong potential for clinical application in the treatment of MM. Additionally, circ_0000190 maintains robust capability for therapeutic suppression of oncogenic processes in other cancers such as osteosarcoma and gastric cancer [50,51]. 

### 4.3. Role of hsa_circ_0007841 in Multiple Myeloma

Song et al. established novel connections between drug resistance and circRNA upregulation in MM. Previous bioinformatic analysis identified differentially expressed circRNAs in MM, leading to selection of hsa_circ_0007841. Significant hsa_circ_0007841 expression was confirmed in the tissues of 86 MM patients. Furthermore, hsa_circ_0007841 expression was found to be closely correlated with drug resistance (DR) and poor prognosis, validating research into its impact on MM. Western blot, qRT-PCR, cell proliferation/viability assays, and half-maximal inhibitory concentration values were employed to evaluate protein/RNA expression and reduction in chemoresistance. Hsa_circ_0007841 was found to be upregulated in U266 doxorubicin resistant cells (U266R) and 8226 doxorubicin resistant cells (8226R). As a result, Song et al. speculated that gene silencing of hsa_circ_0007841 in U266R and 8226R could reduce the half-maximal inhibitory concentration which would indicate reduction in chemoresistance. Further, the mRNA and protein level of ATP-binding cassette transporters G2 increased in doxorubicin resistant cells. Both mRNA and protein levels of ATP-binding cassette transporters have the potential to decrease as hsa_circ_0007841 is silenced. Inhibiting ATP-binding cassette transporter G2 could hinder hsa_circ_0007841 overexpression-induced chemoresistance in U266P and 8226P cells. Further, ATP-binding cassette transporter G2 could reduce differences of half-maximal inhibitory concentration between parent cell lines and drug-resistant cell lines. Collectively, the data distinctly portrayed a new model where hsa_circ_0007841 promotes acquired chemoresistance by upregulating ATP-binding cassette transporters G2, thus presenting a complex basis of chemotherapy in MM [52]. 

## 5. Gastric Cancer

Gastric cancer (GC) is one of the most diagnosed carcinomas (>1,000,000 yearly cases), ranking as the third most common cause of cancer-related deaths, globally [53]. GC risk factors include, but are not limited to, previous or co-infection with *Helicobacter pylori*, presence of precursor lesions, genetics (i.e., polymorphisms), as well as environmental factors [54]. Additional factors, such as geographical location, have been shown to disproportionately impact GC statistics—i.e., higher incidence rates are found in East Asia, Eastern Europe, Central America, and South America, when compared to other regions globally [55]. At time of diagnosis, surgical resection has high success rates for growth control and prevention of spread; however, a great number of personalized decisions are critical for evaluating treatment and resection options. The current diagnostic standard includes a combination of endoscopic sampling and enhanced CT scans. These methods are not plausible for the long-term evaluation of disease progression, contributing to an overall poor patient survival outcome [56]. Further, commonly used markers, such as CEA, CA19-9, CA12-5, and CA72-4 have been used as diagnostic tools, but have been repeatedly shown to have poor prognostic ability and do not aid in detection of early-stage GC [57,58]. Personalized decisions, i.e., decisions regarding disease prevention, risk stratification, and biomarker identification, are viable efforts in combating GC establishment and progression, and are essential for successful screening processes [59]. Thus, the identification of biomarkers that aid in the efficiency of diagnostic screening or actionable targets that can aid in treatment successes would both contribute immensely to earlier stage diagnoses as well as increased survival rates, respectively [56]. The aforementioned diagnostic markers, such as CA72-4, serve as the gold standard of early diagnostic biomarkers—despite poor sensitivity and specificity. To ameliorate this hurdle, circRNAs notably have tissue-specific expression patterns in both solid tissue and plasma of pathologies [60]. Further, recent GC research has enabled the identification of circRNAs that contribute to central carcinogenic mechanisms (i.e., migration, proliferation, and invasion) as well as tumor differentiation and TNM staging [61]. 

### 5.1. Role of hsa_circ_0074362 in Gastric Cancer

In lieu of *replacing* currently used biomarkers for diagnosis and prognosis, circRNAs can, and should, be used alongside established markers to increase specificity and accuracy. For example, hsa_ circ_0074362 was identified by Xie et al. as a downregulated transcript in gastritis and GC tissue but was not proposed as an independent biomarker due to its low sensitivity. However, hsa_ circ_0074362 levels were found to be associated with the aforementioned marker, CA19-9, and LN metastasis [62]. 

### 5.2. Role of circRNA_102231 in Gastric Cancer

A study conducted by Yuan et al. identified differentially expressed circRNA molecules using Gene Expression Omnibus (GEO) datasets (GSE152309, GSE141977, GSE121445) that provided genomic profiling of GC and non-cancerous patient tissues. Amongst the three GEO datasets, circRNA_102231 was identified as the shared, highest expressing, differentially expressed transcript. Further in vitro and in vivo evaluation provided functional mechanisms of circRNA_102231 in GC cells; circRNA_102231 serves as a protein binding scaffold for IRTKS (insulin receptor tyrosine kinase substrate), a known oncoprotein, increasing the half-life (from 2 h to 8 h) of this protein, preventing its degradation. This mechanism, in turn, increased the duration IRTKS exerted its oncogenic behavior. In GC, IRTKS has been previously identified to promote p53 ubiquitination and subsequent degradation [63]. Correspondingly, IRTKS is an additional upregulated transcript in GC patient tissue, and its expression is predicted to be correlated with high circRNA_102231 expression. These data collected by Yuan et al. propose an example of a significant post-transcriptional modifying role for circRNAs [56]. 

### 5.3. Role of circRIMS in Gastric Cancer

Lin et al. evaluated the expression profiles of gastric adenocarcinoma tissues (three T3N3M0; three T3N1M0). The RNA sequencing results identified 207 circRNAs that were differentially expressed between the T3N3M0 and T3N1M0 samples, of which 116 circRNAs were upregulated, with the remaining 91 downregulated. Of these transcripts, a novel circRNA, circRIMS, was identified and was the most upregulated transcript in the comparative cohort. Using miRanda [18], hsa-mirR-148a-5p and hsa-miR-218-5p, two functionally annotated tumor-related miRNAs, were identified as potential targets of circRIMS. Both hsa-miR-148a-5p and hsa-miR-218-5p have been identified previously as tumor suppressors due to their ability to target and downregulate oncogenic transcripts such as PAI-1, ITGA5, SMAD2, and MMP7. CircRIMS was found to act as a molecular sponge and downregulates the expression and availability of both hsa-miR-148a-5p and hsa-miR-218-5p, thereby increasing the expression of their mRNA targets. The pathological upregulation of circRIMS in GC therefore has been demonstrated to drive tumorigenesis by promoting the invasive and migratory propensity of gastric tumor cells. Lin et al. also concluded that circRIMS can serve as a valuable tool for both early detection of GC as well as a concrete guide for clinical diagnosis and precision medicine, hurdles that greatly impede the success of GC diagnosis and treatment [64].

### 5.4. Role of hsa_circ_0000745 in Gastric Cancer

Huang et al. investigated the differential expression patterns of 20 GC tissue samples vs. paired normal tissue and identified three putative circRNAs (hsa_circ_0000745, hsa_circ_0085616, and chr16:30740286-30740893) for evaluation. Hsa_circ_0000745 was found to be of most significance following a predictive circRNA-miRNA network analysis and was downregulated in all 20 GC patient tissues. Further evaluation of hsa_circ_0000745 in 60 GC tissues and 60 plasma samples compared with paired normal tissues identified a clinical correlation between its downregulation and worse pathologies in these patient cohorts, specifically tumor differentiation. Hsa_circ_0000745 was functionally predicted to serve as a miRNA sponge; analysis using sequence complementarity identified an abundance of putative miRNA targets that are known to regulate GC phenotypes, such as hsa-miR-335-3p, hsa-miR-100-3p, and hsa-miR-185-3p, functioning as tumor suppressors that inhibit cell proliferation and regulate chemotherapeutic responsiveness. Further experimental evaluation is warranted to determine the efficacy of hsa_circ_0000745 as a diagnostic marker, and whether these predictive miRNA interactions are bona fide modulators of GC cell phenotypes [65]. 

## 6. Thyroid Cancer

Thyroid cancer (TC) is the most common endocrine malignancy, varying in diagnostic, prognostic, and therapeutic successes. The disparity between these variables is due to several distinct carcinoma subtypes that arise in the thyroid, based on degree of differentiation [66]. Well-differentiated TC represents the vast majority of TC cases and includes both papillary thyroid cancer (PTC) (80% of cases) and follicular thyroid cancer (FTC) (10–20% of cases). PTC and FTC are typically associated with better patient outcomes due to therapeutic responses; however, instances of disease recurrence remain prominent issues. Anaplastic thyroid cancer (ATC) (<2% of cases), a type of undifferentiated TC, is a rapidly growing and lethal cancer [67]. Well-differentiated TC is diagnosed with a typical diagnostic system of stages I–IV, whereas undifferentiated TC is always diagnosed as stage IV metastatic disease, receiving separate diagnostic criteria split into stage IVA, stage IVB, and stage IVC. Diagnostic strategies for TC center primarily on fine needle aspiration; however, this technique is typically ineffective when differentiating FTC phenotypically from PTC. Additionally, the transition from well-differentiated TC to aggressive undifferentiated TC, from a molecularly driven standpoint, is vastly understudied. Further, a common driver mutation across TC subtypes is the BRAF point mutation, or other mutations of its RAS-MEK-ERK signaling cascade [68]. Mutational status of TC alone has not been a successful determinant of prognosis and disease activity; thus, identification of novel biomarkers that aid in subtype differentiation, as well as prognostic evaluation within subtypes, is warranted. There have been several recent studies that have identified a critical role for circRNAs in TC at a multitude of carcinogenic levels.

### 6.1. Role of circFOXM1 in Thyroid Cancer 

A study conducted by Ye et al. identified 1137 differentially expressed circRNA molecules comparing PTC vs. paired normal tissue, including a plethora of novel circRNA transcripts. Several circRNAs were selected for evaluation, specifically, circ_0025033, which was subsequently termed circFOXM1. It was observed that higher levels of circFOXM1 were found in patients with more aggressive disease (i.e., distant metastases) and its expression was directly correlated with increased tumor size. Functional analysis of circFOXM1 in vitro confirmed a significant role in PTC cell growth and proliferation, through the direct targeting and sponging of miR-1179, and the indirect positive regulation of HMBG1 through a ceRNA axis. HMBG1 is a previously identified oncogenic genomic regulator of cancer-associated transcripts [69] and was additionally found to be upregulated along with circFOXM1 in the same patient cohort used for this study. Thus, novel circFOXM1 expression was correlated with increased carcinogenic behavior of PTC in vitro, functioning through a circRNA/miR-1179/HMBG1 ceRNA tumor-promoting regulatory axis [70]. 

### 6.2. Role of hsa_circ_0058124 in Thyroid Cancer

Yao et al. investigated circRNA expression of PTC tissues using a CRISPR RNA (crRNA) microarray. With interest in mechanistic association with PTC tumorigenesis and invasiveness, they additionally screened 13 of these circRNAs using available bioinformatic software analysis. It was found that high expression levels of hsa_circ_0058124 was significantly correlated with a worse prognosis for PTC patients and was therefore further investigated for its impactive role on PTC tumorigenesis. Hsa_circ_0058124 is genomically located on chromosome 2, in the neighborhood of the host gene, Fibronectin 1 (FN1). FN1 is a previously annotated PTC biomarker, and is a significant component of the extracellular matrix, greatly contributing to more aggressive and metastatic cases of PTC. Upregulation of FN1 in aggressive PTC cases was found to be associated with an additional increase in expression of hsa_circ_0058124, located five exon junctions away from the FN1 locus. To understand the cellular mechanisms employed by hsa_circ_0058124, its role as a molecular sponge was investigated. It was found previously by Yao et al. that hsa_circ_0058124 impacted NOTCH3 signaling pathways within the cell and was therefore speculated to do so through a ceRNA axis. Using transcriptomic bioinformatic web tools, i.e., CircInteractome [20] and StarBase [23], and further in vitro assessment using luciferase reporter assays, hsa_circ_0058124 was predicted to bind miR-665 and miR-218-5p. Further, it was reported that miR-218-5p can simultaneously bind hsa_circ_0058124 and Numb, a significant Notch signaling suppressor. Notch signaling, in a multitude of cancer types, functions to prevent initiation and propagation of tumorigenesis and serves as an extensive cell cycle regulator via Mybl2 [71]. The expression level of Numb was detected in PTC patient samples via qRT-PCR and found to be significantly upregulated along with hsa_circ_0058124, confirming a ceRNA axis of regulation between these three transcripts. This was confirmed by the significant upregulation of the NOTCH3 signaling pathway upon suppression of hsa_circ_0058124 expression and upregulation of miR-218-5p expression. Yao et al. concluded that hsa_circ_0058124 was promoting PTC tumorigenesis and progression through the intra-regulatory network that suppresses NOTCH3 signaling [72]. 

### 6.3. Role of circRNA_0057209 in Thyroid Cancer

Bioinformatic analysis conducted by Peng et al. of GEO datasets GSE93522 and GSE40807 was used to identify differentially expressed circRNAs and miRNAs, respectively, in thyroid cancer cohorts. Results of the analysis showed that there were 128 differentially expressed circRNAs, broken down into 108 upregulated and 20 downregulated transcripts in the GSE93522 patient cohort. The selected candidate, circRNA_0057209, was reportedly downregulated in thyroid cancer tissue. Further analysis of differentially expressed miRNAs using GSE40807 was used to attempt to identify potential downstream targets of circRNA_0057209 that were previously predicted using the CircInteractome [20] and circBank [24] databases. At the intersection of these analyses, hsa-miR-183 was identified as a predicted regulated transcript of circRNA_0057209. Further, hsa-miR-183 was found to play a role in thyroid cancer tumorigenesis, by targeting programmed cell death 4 (PDCD4) [73]. Further analysis using Metascape v3.5 was implemented as a prediction method for identifying roles for these transcripts in cancer-related pathways. Of these, the serine/threonine kinase 4 (STK4) gene was a predicted output, which has been previously determined to have a significant role in thyroid cancer metastasis through the Hippo pathway. Using StarBase v2.0, several binding sites of hsa-miR-183 and STK4 were identified, further confirmed by a negative expression correlation in thyroid cancer patient cohorts. STK4 was found to serve as a suppressor of thyroid cancer carcinogenic phenotypes in vitro such as proliferation, migration, and invasion through the induction of apoptotic pathways. Mechanistically, STK4 has the ability to phosphorylate YAP [74], and therefore regulate Hippo signaling pathways. This was further supported by results obtained by Peng et al. in thyroid cancer, where STK4 silencing resulted in a reduction in key proteins involved in cell cycle arrest via Hippo signaling (MST1, LAST1, and LAST2), resulting in the promotion of carcinogenic phenotypes. The in vivo assessment of this mechanism confirmed that the experimental overexpression of circRNA_0057209 was able to reduce tumor volume, a positive effect that was negated when STK4 was experimentally silenced. Therefore, Peng et al. were able to confirm a tumor-suppressive role for circRNA_0057209 and STK4, diminished in thyroid cancer tumor profiles that had significantly reduced expression of both transcripts [75].

## 7. Melanoma

Melanoma is considered the most serious type of skin cancer, causing approximately 75% of skin cancer-related deaths with an incidence of 15–25 per 100,000 individuals [76]. Melanoma arises from the occurrence of genetic mutations in melanocytes, key pigment-producing cells [77], which can be either acquired through sun exposure or inherited. For example, families with melanoma who have germline mutations in *CDKN2A* are well known, whereas the vast majority of sporadic melanomas have mutations in the mitogen-activated protein kinase (MAPK) cascade, which is the pathway with the highest oncogenic and therapeutic relevance for this disease. Early detection of this type of cancer is imperative as effectiveness of treatments decreases in metastatic melanoma resulting in a survival rate of 6–12% once metastatic [78]. Typical treatments include surgical resection, chemotherapy, radiotherapy, photodynamic therapy (PDT), immunotherapy, or targeted therapy depending on tumor location, stage, and genetic profile [77]. For patients with advanced stage melanoma, BRAF inhibitors, MEK inhibitors, and PD-1-specific antibodies have been used to improve patient prognosis [78]. However, only a small subset of metastatic patients have long-term benefits from these current treatments [76], and therefore there is a great need for predictive biomarkers in melanoma patients. 

### 7.1. Role of circ_0001591 in Melanoma

Yin et al. evaluated the expression patterns of 53 melanoma patients, comparing serum samples from both melanoma and healthy controls. They found that the expression levels of circ0001591 were significantly increased in melanoma samples compared to the normal controls. Further, circ0001591 expression was found to significantly correlate with both decreased overall survival as well as disease-free survival when compared to patient samples that had lower expression levels. Experimental evaluation of circ0001591 in vitro using anti-circ0001591 mimics enabled the identification of dynamic roles for this molecule in the regulation of melanoma cell growth. Gene chip analysis revealed a downregulation of miR-431-5p in melanoma patients, and an in vitro model of reduced circ0001591 expression exhibited a marked upregulation of this miRNA, while overexpression of circ0001591 reduced its expression. Additional analysis showed a role for miR-431-5p regulation of both PI3K and ROCK1 expression. Specifically, in melanoma, miR-431-5p regulates apoptotic induction through the direct interaction with ROCK1; therefore, its decreased expression because of circ0001591 removes the apoptotic induction carried out by this regulatory pathway. The overexpression of miR-431-5p suppressed PI3K expression and Akt phosphorylation through the targeted inhibition of ROCK1 protein expression. On the contrary, overexpression of circ0001591 was found to induce both PI3K expression and Akt phosphorylation by promoting ROCK1 expression. This premise further suggested an interactive regulatory role and a circ0001591/miR-431-5p/ROCK1 regulatory axis. Identification of this novel interactive network significantly aids in the classification of malignant melanoma and can aid in its unknown etiological establishment [79]. 

### 7.2. Role of hsa_circ_0025039 in Melanoma

Bian et al. evaluated the transcriptomic expression patterns of malignant melanoma. Using a microarray, they identified six differentially expressed circRNA transcripts in malignant melanoma (43 melanoma tumors vs. 18 paired normal skin tissue). It was identified that hsa_circ_0025039, a circRNA that is derived from the exons of FOXM2, was an overexpressed transcript, as well as the most significantly dysregulated of the sample cohort. Evaluation in vitro demonstrated a significant role for hsa_circ_0025039 in melanoma cell proliferation, invasiveness, and glucose metabolism, all cellular events that drive aggressive tumor establishment and behavior. To understand the molecular orchestration of hsa_circ_0025039 on these phenotypes, a dual-luciferase reporter assay identified a putative miRNA target—miR-198. In accordance with this notion, miR-198 expression was found to be significantly decreased in melanoma tissue samples compared to the healthy controls. As stated in the introduction, understanding genomic differences between cancerous and non-cancerous tissue serves as an invaluable source of information; however, differences in genomic expression within the same cancer with differing parameters (i.e., tumor stage, therapeutic response, survival analysis) is a particularly advantageous source of information. Bian et al. further reported that miR-198 expression was further downregulated in primary melanoma tissues, corresponding with the increase in hsa_circ_0025039 expression. TargetScanHuman 8.0 prediction software identified CDK4 as an miR-198 target, with significant putative binding sites. Experimental addition of miR-198 mimics in vitro significantly reduced the expression of CDK4 at the protein level when compared to the control, confirming a negative interaction between these two transcripts. In accordance with this, high CDK4 expression was also observed in the melanoma tissue compared to the healthy skin tissue and was similarly also increased in expression in primary melanoma when compared to metastatic melanoma—the pattern previously observed with hsa_circ_0025039 and miR-198. Silencing of hsa_circ_0025039 expression significantly reduced tumor growth in vivo, corresponding to the IHC analysis of tissue which showed a marked decrease in CDK4 expression compared to controls. Bian et al. not only identified hsa_circ_0025039 as a putative marker for metastatic melanoma, but also further contributed to the identification of differential expression patterns of varyingly progressed tumor tissues of the same cancer type [80].

### 7.3. Role of circ_0084043 in Melanoma

A study evaluated the transcriptomic expression patterns of 30 primary cutaneous melanoma patient samples compared to non-cancerous skin tissue of each patient. A previously evaluated circRNA, circ_0084043, had been found to serve as a significant contributor to malignant melanoma progression [81], and was evaluated further by Chen et al. in this study. It was detected that expression patterns of circ_0084043 was similarly upregulated in this patient cohort. As mentioned, the ceRNA axis of non-coding RNA molecules is a significant contributor to genomic instability and carcinogenic induction. TRIB2 (Tribbles homolog 2) has been previously described as an upregulated transcript in melanoma cells [82], and its mRNA was additionally upregulated and its expression was correlated with circ_0084043. Pearson correlation analysis of transcriptomic expression patterns confirmed a strong correlation between circ_0084043 and TRIB2. Evaluation of function in vitro using siRNAs against circ_0084043 identified an impactive role on melanoma cell proliferation and evasion of apoptotic induction—significantly impacting overall cell survival. Further, evaluation of TRIB2 in these melanoma cell lines confirmed its expression promotes cell survival in a similar fashion. To connect a regulatory axis between circ_0084043 and TRIB2, several known miRNAs that negatively regulate TRIB2 were evaluated. It was found that MiR-429 serves as a direct regulator of TRIB2, and binding sites for miR-429 were present on circ_0084043—a key indicator of molecular sponging action. These connections are confirmed by the expression patterns and functional readouts of their genomic functions. High circ_0084043 and TRIB2 expression, with low miR-429 expression, in malignant melanoma, elucidates interactive regulatory connections between these three RNA molecules. β-catenin, a well-studied regulator of carcinogenic orchestration, was identified as a novel target of this circ_0084043/miR-429/TRIB2 axis, and that the sponging of miR-429 via circ_0084043 and upregulation of TRIB2 leads to the inactivation of the Wnt/β-catenin pathway. The observation of this interactive axis in malignant melanoma suggests a putative approach for identifying mechanisms that drive the malignant progression of primary melanoma, thus supplying plausible actionable targets [83]. 

## 8. Breast Cancer

Breast cancer (BCa) is one of the most common malignancies developed in women, with a diagnostic statistic of greater than 1.5 million women each year [84]. Primary metastatic sites for BCa include the bones, lungs, and the brain, significantly contributing to the hurdles faced when developing effective treatments. Identification of BCa prior to metastatic spread is key to increasing treatment efficacy and better patient outcomes. Routine examinations, such as mammography and MRI, have significantly contributed to a higher 5-year survival rate, which is now above 80% [85], despite the rapid increase in incidence yearly. From a molecular standpoint, there have been a significant portion of putative genes that drive BCa initiation and progression. For example, breast cancer associated gene 1 and 2 (*BRCA1* and *BRCA2*) are the most high-risk oncogenes identified thus far. *BRCA1* and *BRCA2* functionally encode tumor suppressor proteins that drive cell cycle immortality and extensive genomic instability [86]. The *HER2* gene encodes for the HER2 protein, which is a significant growth factor that propagates and drives BCa. Hormone status classification of BCa is a key feature of diagnosis. Estrogen receptor (ER)-positive cancers contain cells that respond to estrogen signaling, progesterone receptor (PR)-positive cancers respond to progesterone signaling, and cancer subtypes that respond to neither estrogen nor progesterone are deemed hormone receptor (HR)-negative. BCa cases that have the lowest survival are those that lack ER, PR, and HER2, and are collectively referred to as triple-negative breast cancer (TNBC). A significant portion of BCa treatment has focused on hormonal therapy, specifically blocking specific hormone expression, thus blocking receptor signaling. These treatments are futile for BCa types that are HR-negative; however, there has been progress in the study of circRNA molecule expression in TNBC—specifically, circRNAs that contribute to aggressiveness and therapeutic resistance (i.e., high circEPSTI1 correlated with poor TNBC prognosis) [87]. Further, within TNBC cases, hsa_circ_0019853 was identified as an independent risk factor for TNBC prognosis, and its high expression was significantly associated with decreased survival [88]. Additional actionable targets are warranted, as well as the molecular understanding of genetic and epigenetic factors that differ between these types. Furthermore, it is reported that 20–25% of inherited BCa is associated with mutations of these genes, which have significantly increased the rate of early detection by result of genetic testing prior to cancer onset [89]. Identification of both early BCa development markers and actionable molecular targets will continue to contribute significantly to better therapeutic options, significantly increasing survival rates. Indebted to genetic sequencing and evaluation, there have been increasing numbers of circRNAs identified as differentially expressed transcripts between non-cancerous and BCa tissue, as well as heterogeneous expression amongst varying BCa types and stages, significantly correlating expression patterns with carcinogenic processes and clinical characteristics [90]. Studies conducted by Rao et al. identified hsa_circ_005046 and hsa_circ_0001791 as significantly upregulated transcripts in early-stage BCa, specifically correlating with high early diagnostic predictive value [91]. On the contrary, circDENND4C was found to be associated with advanced BCa and degree of metastasis [92]. Due to the extensive heterogeneity of BCa, identification of uniform biomarkers can contribute significantly to feasible diagnosis and predictive measures [93]. 

### 8.1. Role of hsa_circ_0001785 in Breast Cancer 

Expression profiling of circulating circRNAs via a microarray in plasma samples from BCa patients was compared to non-cancerous plasma, detecting a total of 41 that were differentially expressed. Yin et al. evaluated the expression patterns of five BCa patients vs. healthy matches via circRNA microarray, identifying 19 upregulated and two downregulated circRNA transcripts. Amongst these, hsa_circ_0001785, an upregulated transcript, was selected based on its significant correlation with diagnostic value, histologic grade, and distant metastasis, when compared to other identified dysregulated circRNAs. A secondary expanded cohort size was evaluated (57 BCa patients) and the diagnostic value of hsa_circ_0001785 was retained, excluding relation to other key BCa contributing factors such as age, lymph node invasion, and hormone receptor status. On continued evaluation of BCa patient plasma, comparing preoperative and postoperative cases, hsa_circ_0001785 expression was found to be significantly lower in postoperative cases, strongly suggesting its role in early detection and initial screening processes [94]. Liu et al. further reported that hsa_circ_001783 was expressed in greater abundance in TNBCs when compared to other HER2+ BCa types. Knockdown of hsa_circ_0001785 suppressed carcinogenesis of BCa cells in vitro via sponging interactions with miR-200c-3p [95]. 

### 8.2. Role of hsa_circ_001783 in Breast Cancer

Liu et al. proposed a bioinformatic analysis strategy to aid in molecular detection of dysregulated noncoding RNAs, specifically to identify ceRNA interactions between circRNAs and miRNAs. A circRNA–miRNA-BCa database was successfully constructed through this study, enabling a feasible evaluation of an abundance of expression markers using the expression profiling of patients in conjunction with sequence complementarity predictions. The strategy of their in silico analysis implemented various predictive algorithms to identify miRNA binding sites. These sites were then predicted to interact with circRNAs using StarBasev2.0 [23]. Analysis using NCBI GEO datasets enabled identification of differentially expressed genes. Expression profiling of 136 BCa samples was compared in conjunction with 50 BCa samples from patients prior to therapy. It was found that hsa_circ_001783 was increased in BCa tissue samples, with an expression correlation with poor clinical characteristics and outcomes (i.e., increased metastasis). In this example, hsa_circ_001783 expression correlated with ER and PR status, as well as molecular subtype (TNBC vs. non-TNBC)—excluding expression correlation with HER2 status, histological grade, and age/menopause. An in vitro assessment of hsa_circ_001783 function was conducted, and it was determined that its expression reduction ameliorated central carcinogenic mechanisms of BCa cell line MDA-MB-231. The molecular mitigation of these processes was found to be through direct interaction with miR-200c-3p, through a ceRNA axis. Moreover, miR-200c-3p has been previously reported to exert biological roles in various malignancies [96]. Liu et al. identified *ZEB1*, *ZEB2*, and *ETS1* as the molecular targets of miR-200-3p in BCa. Confirming an interactive “sponging” role between hsa_circ_001783, miR-200-3p, and *ZEB1*, *ZEB2*, and *ETS1*, expression values were directly and inversely correlated according to traditional ceRNA regulation. High hsa_circ_001783 expression correlated with decreased miR-200-3p expression and high *ZEB1*, *ZEB2*, and *ETS1* expression in the BCa patient cohort. This study’s comprehensive analysis using both in silico and in vitro assessment has served as a significant example of evaluation tools for identifying a plethora of biologically relevant transcripts [97]. 

### 8.3. Role of circEPSTI1 in Breast Cancer

Frequently mutated pathways in BCa include the proliferative and growth promoting signal transduction cascade orchestrated by the PI3K/Akt/mTOR axis. Previous work reported significant contributing roles for circRNAs and the modulation of this pathway, leading to increased survival of cancer cells and poorer outcomes in patients. Zhang et al. identified a highly expressed circRNA, circEPSTI1, in HER2-positive BCa tissue, and found that it regulates and preferentially activates PI3K/Akt signaling. Comparing 20 HER2-positive and non-cancerous adjacent breast tissue samples, circEPSTI1 was upregulated in all samples. Evaluation of circEPSTI1 molecular orchestration revealed a significant contribution to cell invasion, proliferation, and migration. Further molecular analysis identified a role for circEPSTI1 in the cytoplasm of BCa cells, serving as a molecular sponge. The predictive algorithms used suggested extensive sequence complementarity sites for miR-145, a transcript that is significantly decreased in the HER2-positive patient cohort and corresponding cell lines. Restoration of carcinogenic phenotypes in si-circEPSTI1 cell lines occurred when miR-145 was inhibited, suggesting a correlation between the cellular effects exerted by circEPSTI1 through this proposed ceRNA axis. The protein ErbB3 has been found to confer resistance to a multitude of therapeutics, such as tyrosine kinase inhibitors, ErbB2 biologics, and hormone therapy—key approaches for treatment of BCa. In ER-positive BCa, ErbB3 is overexpressed after fulvestrant treatment and drives hormonal therapeutic resistance via MAPK/ERK pathway activation [98]. Interestingly, ErbB3 was identified by Zhang et al. as a predicted target of miR-145, and experimental evaluation in vitro confirmed that ErbB3 expression was in fact inhibited by miR-145 expression. A previous study reported a direct correlation between increasing miR-145 expression and chemotherapeutic efficacy, further suggesting that instances of its dysregulated underexpression via circEPSTI1 overexpression can be detrimental to treatment responses. It was reported that miR-145 overexpression had the ability to induce the accumulation of doxorubicin by suppressing MRP1 in BCa cells, significantly increasing therapeutic responses [99]. 

## 9. Gynecologic Cancers

Gynecological malignancies, namely, ovarian, cervical, and endometrial cancer, contribute a significant portion to global cancer statistics. Further, advanced and recurrent gynecological malignancies are increasing in incidence, and are significantly associated with poor prognoses due to the extensive lack of effective therapeutic options [100]. For perspective, with the global incidence rate of 1,000,000, the mortality rate is 500,000 annually, representing approximately half of the diagnosed cases. Despite the differences in driver mutations that persist between each gynecological cancer, they are all equally lacking in biomarker identification and therapeutic approaches [100]. For example, the use of anti-angiogenic agents to treat gynecologic cancers has been shown to be significantly prone to resistance over time, additionally presenting with an extensive lack of molecular characterization of said treatment and resistance. The process of angiogenesis is a significant contributor to the development of these cancer types, due to its extensive role in physiological processes that regulate the female reproductive cycle, hence its proposition as a potential therapeutic target. Angiogenic processes within the female reproductive tract serve as significant drivers of malignant pathogenesis by de-regulating regular physiological processes. It is therefore critical that novel angiogenic markers are identified in order to contribute to prognostic value and therapeutic adjuvant success [101]. It has been reported that circRNAs have been serving as contributors to chemotherapeutic resistance in these cancer types—highlighting their potential as treatment targets and/or biomarkers for prognosis/treatment response. 

## 10. Ovarian Cancer

Ovarian cancer (OC) is the most significant contributor to the cause of gynecologic malignancy death, the seventh most common cancer diagnosed in women, and represents approximately 3.3% of diagnosed female cancers [102,103]. The five-year survival rate of 30% in OC is an extensive reflection of the lack of effective diagnostic and therapeutic strategies. Three main forms of OC persist: serous, endometroid, and mucinous—or combinations of each. All forms present with unique molecular characteristics and individual response rates to chemotherapy. Several comprehensive bioinformatic analyses were published that have statistically correlated circRNA expression with patient survival, clinicopathological features, and therapeutic resistance [104,105]. 

Other circRNAs associated with pathological grade of OC include circ_0072995 [106], circ-FAM53B [107], circITCH [108], and circRNA ABCB10 [109].

### 10.1. Role of hsa_circ_0013958 in Ovarian Cancer

A study conducted by Pei et al. evaluated the expression levels of hsa_circ_0013958 in 45 OC tissue samples for identification of clinicopathological feature association. It was reported that hsa_circ_0013958 expression was increased in OC tissues and OC cell lines, and significantly correlated with lymph node metastasis. A unique feature of OC, collectively, is the ability for the cancer cells to co-express epithelial and mesenchymal markers—key drivers of Epithelial-to-Mesenchymal transition (EMT), a process that drives migratory and metastatic spread [110]. The in vitro assessment of hsa_circ_0013958 function confirmed a role for this molecule in EMT, significantly impacting migration and invasion of OC cells in culture. Functionally, hsa_circ_0013958, when silenced, increased E-cadherin and downregulated vimentin, supporting a less carcinogenic and mesenchymal phenotype by modulating EMT markers. It was further shown that a reduction in hsa_circ_0013958 expression increased apoptotic induction through the upregulation of Bax and downregulation of Bcl-2 [111]. Another circRNA identified as a significant contributor to OC EMT, circRNA_100395, was found to promote proliferation and metastatic mechanisms through regulation of miR-1228 and p53 interaction. When underexpressed, circRNA_100395 increases miR-1228 and serves as an indirect regulator of p53, by removing miR-1228 inhibition. Thus Li et al. contributed significantly to the identification of other EMT drivers in OC [112]. 

### 10.2. CircRNA Molecule Expression Drives Therapeutic Resistance in Ovarian Cancer

It was reported that hsa_circ_0078607, a novel circRNA, is significantly downregulated in OC, and was the subject of a functional analysis performed by Zhang et al. Structurally, hsa_circ_0078607 is derived from an exonic back-splicing event of the SLC22A3 gene, an extensively studied solute transporter in other cancer types [113,114,115]. This study identified hsa_circ_0078607 as an miRNA sponge for miR-518-5p, causing the elevation of pro-apoptotic Fas mRNA expression [116]. 

As mentioned, chemoresistance remains a significant problem in OC. Comparative genomic analysis was carried out of cisplatin-resistant vs. responsive OC tissue. Underexpressed circRNA, Cdr1, was identified as a significant contributor to cisplatin resistance through its downstream consequences of downregulation. Low Cdr1 expression was found to promote miR-1270 expression, which in turn regulates and decreases SCAI (suppressor of cancer cell invasion) expression. Functional evaluation in vitro confirmed that overexpression of Cdr1 inhibited proliferation in cisplatin-treated OC cells—a result confirmed also by the inhibition of miR-1270 [117]. A separate study showed that circRNA FGFR3 is overexpressed in OC and significantly contributes to poor survival, and mechanistically promotes EMT. Zhou et al. reported a novel ceRNA axis of regulation between FGFR3/miR-29a-3p/E2F1, which was found to serve as the significant driving force of EMT [118]. Paclitaxel resistance in OC also remains a prominent issue. CircTNPO3 was identified by Xia et al. for its extensive role in paclitaxel resistance mechanisms in OC, and is significantly overexpressed in OC tissue compared to paired adjacent normal tissue. Further, this overexpression is correlated with decreased survival of OC patients. Functional analysis following paclitaxel treatment confirmed a novel regulatory interaction of circTNPO3 with miR-1299, thus removing negative regulation of NEK2 (NIMA-related kinase 2), an extensive cell cycle regulator and putative oncogene in OC [119]. In accordance with this notion, circCELSR1 was identified by Zhang et al. for its contribution to paclitaxel resistance in OC through a different regulatory axis with miR-1252 and FOXR2 (forkhead box 2) [120]. 

## 11. Endometrial Cancer

Endometrial cancer (EC), arising in the endometrial layer of the uterus, is increasing in incidence globally, with risk of disease recurrence remaining a prominent issue. The standard treatment regimen includes primary hysterectomy, as the disease is typically identified while still confined to the uterus. However, histological factors such as subtype, lymphovascular expansion, disease stage, and other co-morbidities significantly impact therapeutic modalities used for treatment. Beyond histomorphology, molecular classification of EC tumors has implicated specific genetic alterations in informing prognosis and treatment. General mutations persist depending on stage and subtype of EC. Type I EC encompasses mostly mutations in the PI3KCA pathway, and less commonly, *KRAS* mutations. Type II EC represents a broad range of subtypes, with heterogenous molecular and genomic components; however, these mostly consist of *TP53* mutations [121]. In both Type I and Type II EC, PTEN genetic alterations are highly prevalent [122]. Type I ECs typically have a good prognosis, while Type II ECs do not, owing to their advanced stage and tendency for disease recurrence [123]. With that, there is a critical unmet need to develop better diagnostic and treatment strategies. Few studies have evaluated the function of circRNAs in EC pathology, but their potential value has been speculated on. RNA sequencing evaluation of EC tissue and matched normal endometrial tissue of six patients resulted in 120 differentially expressed circRNAs, mainly highlighting their lower expression in EC tissues [124]. The functionality in EC remains to be elucidated but they may serve as diagnostic and prognostic biomarkers in EC. 

### Role of hsa_circ_0001610 and hsa_circ_0002577 in Endometrial Cancer

The establishment of dysregulation of circRNA expression in EC prompted the investigation of those with high-risk EC. Screening of two patients’ EC tumor tissue and matched adjacent normal tissue identified hsa_circ_0001610 as one of the top three most upregulated circRNAs. Further qPCR analysis demonstrated that this circRNA was also upregulated in low-grade EC tumors, just to a lower extent [125]. The expression of this circRNA, also known as circTNFRSF21, was confirmed to be higher in EC tumor tissue, and furthermore its expression was associated with rapid EC cell growth and proliferation, and xenograft tumor formation in mice. The elucidated mechanism includes the sponging of miR-1227 in EC cells which promotes MAPK13/ATF2 signaling [126]. Further characterization of this circRNA was performed, implicating its role in cell migration, invasion, and apoptosis. The proposed mechanism includes hsa_circ_0001610 modulating STAT3 expression through miR-646 sponging [127]. To extend the functional phenotypes elicited by hsa_circ_001610 expression, evaluation of circRNAs implicated in radioresistance, a prominent clinical obstacle in EC treatment, was performed, highlighting the reduction in hsa_circ_0001610 in exosomes of radiosensitive EC cells. The mechanism revealed upregulation of cyclin B1, a driver of radioresistance in several cancer types, through miR-139-5p [128]. Therefore, this research provides new targets for high-risk EC. 

Molecular characterization of EC tumors compared to normal endometrial tissue revealed the potential modulation of hsa_circ_0002577 in the progression of EC. The significant upregulation further correlated to advanced stage, lymph node metastasis, and poor survival. Functionally, hsa_circ_0002577 demonstrated roles in proliferation, migration, and invasion in vitro with effects on tumor growth in vivo, attributed to the sponging of miR-197 and downstream targeting of the CTNNC1/Wnt/β- catenin axis [129]. Confirmation of its upregulated expression in tumor samples and in vitro and in vivo phenotypic effects, further characterization of hsa_circ_0002577 demonstrated its role in accelerating progression of EC via activation of the IGF1R/PI3K/Akt signaling pathway through sponging of miR-625-5p [130]. This circRNA, also known as circWDR26, was evaluated in EC cells and demonstrated, in addition to the previously mentioned functions, inhibition of apoptosis. Mechanistically, this circRNA sponged miR-212-3p that acts through MSH2 to promote EC progression, which further correlated with increased risk and poor prognosis of EC [131]. This research suggests the potential use of this circRNA in treatment of EC. 

## 12. Cervical Cancer

Cervical cancer (CC) is one of the most prevalent malignancies, specifically presenting in young women, reaching a mortality rate of 270,000 [132]. Ninety-five percent of cases are caused by infections with carcinogenic human papilloma viruses (HPVs), and while prophylactic vaccines are available, vaccine compliance is low. There is a critical need for innovational strategies of diagnosis, prognosis, treatment, and management of CC due to the high propensity for recurrence, metastasis, and therapeutic resistance, contributing to its poor survival. Over seventy percent of CCs exhibit somatic alterations in signaling pathways PI3K/Akt-MAPK and TGFB, which are mutually exclusive [133]. However, even with underlying molecular mechanisms of CCs, therapeutic effects are not ideal for recurrent or late-stage disease and novel therapeutic targets are needed. Differential expression of circRNAs in CC have demonstrated potential roles in biomarkers of early detection, prognosis, and personalized treatment targets. Among the plethora of circRNAs differentially expressed in patient tissues, only a fraction of them have been validated functionally in diseases, including CC. Increasing evidence has supported their role in CC tumorigenesis through their effected protein targets, including, but not limited to, hsa_circ_0018289 and hsa_circ_0023404 [134]. 

### Role of hsa_circ_0023404 in Cervical Cancer

When comparing circRNA expression of CC tissues to adjacent normal tissues, hsa_circ_0023404 was significantly upregulated, and its expression was associated with lymph node metastasis, larger tumor size, advanced clinical stage, and decreased survival. Inhibition of hsa_circ_0023404 in vitro demonstrated its role in proliferation, cell cycle progression, cell migration, and cell invasion. Mechanistically, this circRNA acts as a sponge of miR-136, which targets TFCP2, that activates the YAP signaling pathway [133]. To establish a mechanism whereby hsa_circ_0023404 regulates metastasis and chemoresistance of CC, inhibition of its expression attenuated invasion of CC cells and lymphatic vessel formation of endothelial cells through direct interaction with miR-5047. It was evident that sponging of miR-5047 both increased VEGFA levels and inhibited autophagy, mediating cancer metastasis and chemoresistance in CC [134]. In further assessment of the effect of this circRNA on malignant characteristics of CC, in vitro hsa_circ_0023404 sponged miR-636 which allowed for increased expression of protein coding gene target CYP2S1. This hsa_circ_0023494/miR-636/CYPS21 axis resulted in downstream malignant phenotypes including increased proliferation, migration, invasion, and decrease in apoptosis [135]. Together, these data suggest hsa_circ_0023404 as a potential target in CC therapy. (Table 2).

## 13. Conclusions

The initiation, propagation, and sustainability of carcinogenic processes have been attributed in great part to genomic instability. Mutations in cellular regulators have been a key source of driving cellular transformation and metastatic propensity, whilst causing treatment refraction and unsuccessful tackling of target proteins. The non-coding portion of the genome represents the vast majority; non-coding RNAs have shaped a new understanding of cellular energy expenditure in both physiological and pathological processes. RNA molecules are often disregarded as unstable molecules when compared to their genomic counterparts; however, their function within the cell has been shown to exceed those of proteins and other functional molecules. CircRNA molecules are extremely stable transcriptomic elements, as they consist of closed 3′ and 5′ ends, and are highly resistant to nuclease degradation. Functionally, circRNAs have been identified as extensive molecular regulators, specifically in their function as a “molecular sponge” negatively regulating miRNA expression and in turn positively regulating mRNA expression. A multitude of studies, including representative examples in this review, have shown that dysregulated expression patterns of circRNAs can orchestrate and promote carcinogenic behavior and confer treatment resistance. The consequence of circRNA expression patterns is shown in the bona fide function of the miRNAs and the mRNAs they regulate. A key challenge faced in cancerous disease is the lack of biomarkers that can aid in early identification/diagnosis or prognostic prediction, significantly influencing treatment plans and patient outcomes. Identifying gene expression patterns that differ between cancerous and non-cancerous tissue, as well as patterns that differ within a cancer type (i.e., treated vs. non-treated; early stage vs. late stage), is the key to diagnostic and therapeutic advances. Bioinformatic analysis of patient data has significantly opened the doors to the identification of molecules that may have biological relevance based on their differential expression. This premise has enabled feasible selection of gene candidates for study and will continue to contribute to the characterization and classification of unknown biomolecules driving carcinogenic patterns. Stable, molecular orchestrators, such as circRNAs, have already begun to elucidate evident and underlying mechanisms employed by cancer cells to promote their survival, serving as a significant contribution to both RNA and cancer biology. 

## Figures and Tables

**Figure 1 biomolecules-14-00384-f001:**
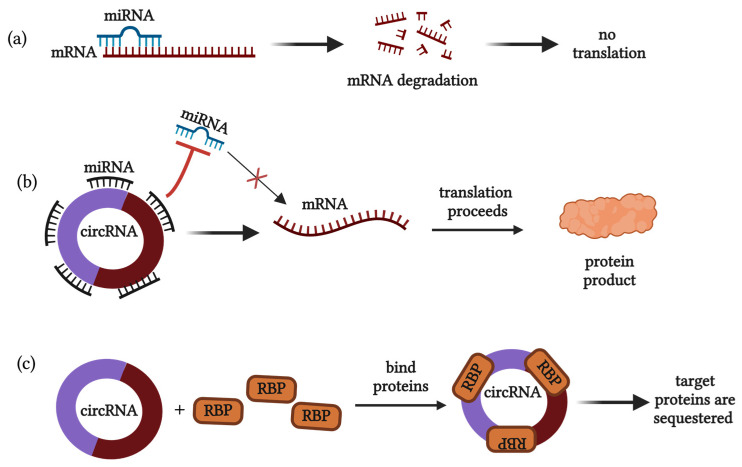
Physiological competing endogenous RNA (ceRNA) axis of circRNAs. (**a**) MiRNAs bind and degrade target mRNAs and block sequence-specific translation from occurring, decreasing protein production; (**b**) CircRNAs contain sequence-complementary miRNA binding sites that sponge miRNAs, resulting in target mRNA translation and protein production; (**c**) CircRNAs utilize protein-binding domains to sequester RNA-binding proteins [Figure created with BioRender (https://www.biorender.com/)].

**Figure 2 biomolecules-14-00384-f002:**
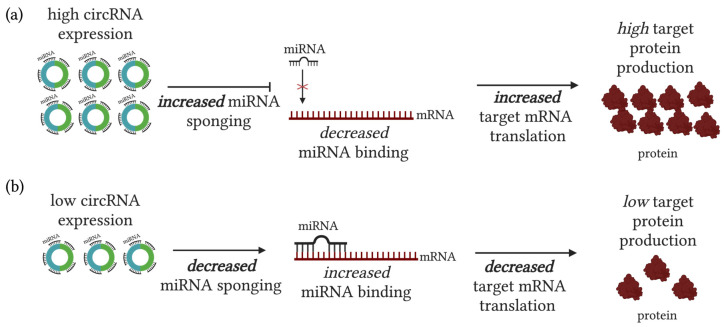
Pathological competing endogenous RNA axis of circRNAs. (**a**) Upregulated circRNA expression increases miRNA sponging and thus decreases miRNA binding of target mRNAs, resulting in dysregulated increase in target mRNA translation and protein production; (**b**) Downregulated circRNA expression decreases miRNA sponging and thus increases miRNA binding of target mRNAs, resulting in target mRNA degradation and decreased protein production [Figure created with BioRender (https://www.biorender.com/)].

**Figure 3 biomolecules-14-00384-f003:**
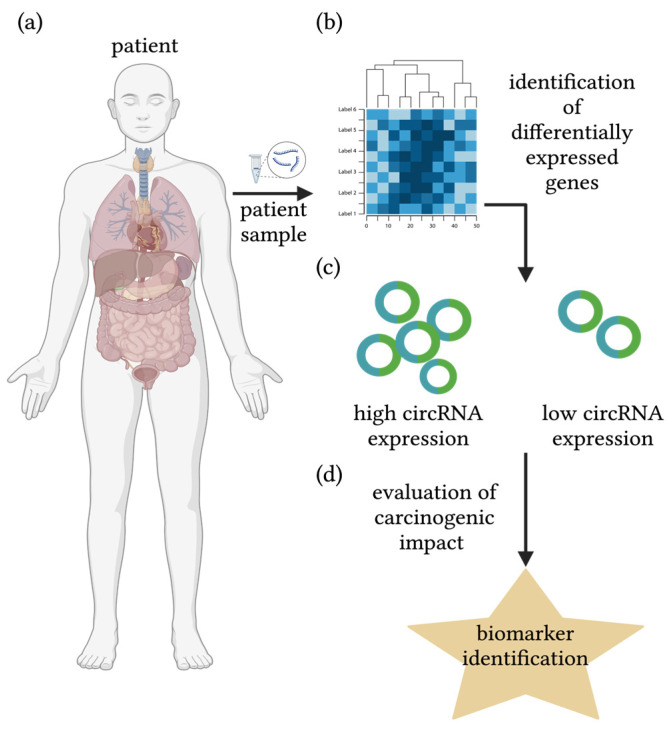
Use of circRNA expression patterns as diagnostic, prognostic, and therapeutic targets. (**a**) Patient samples extracted and sequenced; (**b**) Bioinformatic analysis of genomic sequences that differ between selected groups (i.e., cancerous vs. non-cancerous; early stage vs. late stage); (**c**) Identification of circRNA molecules that are up- or downregulated in sample set; (**d**) Identify any correlation with survival, metastasis, TNM stage, tumor size, etc., and evaluate function of circRNA molecule in vitro and in vivo to identify role as a putative biomarker [Figure created with BioRender (https://www.biorender.com/)].

**Table 1 biomolecules-14-00384-t001:** List of databases available for in silico analysis of circRNA interactome.

Database	Main Identification	Link
miRanda	miRNA target scanner for predicting mRNA targets for miRNAs	https://bioweb.pasteur.fr/packages/pack@miRanda@3.3a (accessed on 15 December 2023) [19]
miRnet	miRNA interaction database for analysis of miRNA–target interactions	https://www.mirnet.ca/ (accessed on 15 December 2023) [20]
CircInteractome	miRNA predictor for target circRNAbinding sites for RBPs and miRNAs	https://circinteractome.nia.nih.gov/ (accessed on 15 December 2023) [21]
miRTarBase	Database of collected miRNA–target interactions using literature surveillance	https://mirtarbase.cuhk.edu.cn/~miRTarBase/miRTarBase_2022/php/index.php (accessed on 15 December 2023) [22]
ENCORI/ StarBase	Analysis of RNA–RNA and RNA–protein interactions and pan-cancer analyses with RNA sequencing data	https://rnasysu.com/encori/index.php (accessed on 15 December 2023) [23,24]
circBank	Database of miRNA binding predictions/assessment of circRNA protein coding potential	http://www.circbank.cn/ (accessed on 15 December 2023) [25]
DIANA-TarBase	Computational model for predicting miRNA and protein coding gene interaction through expression regulation	https://dianalab.e-ce.uth.gr/tarbasev9/interactions (accessed on 15 December 2023) [26]
CircNet 2.0	Database for tissue-specific circRNA expression profiles and circRNA–miRNA interactive networks	https://awi.cuhk.edu.cn/~CircNet/php/index.php (accessed on 15 December 2023) [27]
circR2Disease v2.0	Database of reported circRNAs identified in over 100 pathologies	http://bioinfo.snnu.edu.cn/CircR2Disease_v2.0/ (accessed on 15 December 2023) [28]

**Table 2 biomolecules-14-00384-t002:** Reviewed circRNA molecules.

CircRNA	Cancer Type	Sample Type	Prognostic Value
circMYBL2	Acute Myeloid Leukemia	Plasma	Therapeutic target
hsa_circ_0075451	Acute Myeloid Leukemia	Bone Marrow	Diagnostic predictor
hsa_circ_004277	Acute Myeloid Leukemia	Bone Marrow	Diagnostic marker
circCHEK1_246aa	Multiple Myeloma	Bone Marrow	Therapeutic target
hsa_circ_0007841	Multiple Myeloma	Tissue	Biomarker
circRNA_102231	Gastric Cancer	Tissue	Biomarker
circRIMS	Gastric Cancer	Tissue	Early detection/diagnosis
hsa_circ_0000745	Gastric Cancer	Tissue/plasma	Diagnostic marker
circFOXM1	Papillary Thyroid Cancer	Tissue	Biomarker
hsa_circ_0058124	Papillary Thyroid Cancer	Tissue	Biomarker
circRNA_0057209	Thyroid Cancer	Tissue	Tumor suppressor
circ_0001591	Melanoma	Serum	Biomarker
hsa_circ_0025039	Melanoma	Tissue	Biomarker
circ_0084043	Melanoma	Tissue	Therapeutic target
circ_0084043	Melanoma	Tissue	Therapeutic target
hsa_circ_0001785	Breast Cancer	Plasma	Early detection/screening
hsa_circ_001783	Breast Cancer	Plasma	Biomarker
circEPSTI1	HER2^+^ Breast Cancer	Tissue	Biomarker
hsa_circ_0013958	Ovarian Cancer	Tissue	Biomarker
hsa_circ_0078607	Ovarian Cancer	Tissue	Biomarker
circRNA FGFR3	Ovarian Cancer	Tissue	Biomarker
circTNPO3	Ovarian Cancer	Tissue	Biomarker
circCELSR1	Ovarian Cancer	Tissue	Biomarker
hsa_circ_0001610	Endometrial Cancer	Tissue	Therapeutic target
hsa_circ_0002577	Endometrial Cancer	Tissue	Therapeutic adjuvant

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
