# Peer review of "Dysregulated Expression Patterns of Circular RNAs in Cancer: Uncovering Molecular Mechanisms and Biomarker Potential"

_biomolecules, 2024, doi:10.3390/biom14040384_

Round 1

Reviewer 1 Report

Comments and Suggestions for Authors

This manuscript represents a review on the potential use of circRNAs as cancer biomarkers, and the authors effort in this direction must be appreciated. However, there are several concerns that should be addressed.

Major comments

A section dedicated to the most commonly techniques employed for the study of circRNAs as cancer biomarkers should be included.

The growing area of circRNAs in cancer as potential biomarkers encompasses several additional valuable results. The authors should consider mentioning some more of these studies in the manuscript. For gastric and breast cancers see doi: 10.3389/fgene.2022.1037120 and doi: 10.3390/biomedicines10030725, respectively. In the melanoma section circ_0084043, circ_0002770, circ_0020710 should be discussed.

CircRNAs described in the manuscript should be listed in a Table that also includes the tumor type, the sample description (i.e. tissue and/or plasma), the prognostic value, the expression status and the function (if known).   

There are some missing references. For example, Introduction, line 38 “In the last decade, a plethora of data have supported a significant role for ncRNAs in the development of many pathological conditions, especially cancer (REF)”, and line 60 “It has been described in various studies in the literature that circRNAs are actively contributing to carcinogenic propagation and progression (to add other REF; i.e. doi: 10.1002/1878-0261.13034)”.

Author Response

Reviewer #1:

A section dedicated to the most commonly techniques employed for the study of circRNAs as cancer biomarkers should be included.

  • We agree that this section is important and therefore it was added in our revision. The paper, in majority, focuses on the molecular roles of circRNAs and their basic science evaluation, and we agree that a section of methodology used is of significant value. This section was added following the introduction.

The growing area of circRNAs in cancer as potential biomarkers encompasses several additional valuable results. The authors should consider mentioning some more of these studies in the manuscript. For gastric and breast cancers see doi: 10.3389/fgene.2022.1037120 and doi: 10.3390/biomedicines10030725, respectively. In the melanoma section circ_0084043, circ_0002770, circ_0020710 should be discussed.

  • These additional study references were thoroughly reviewed by our authors and we agree that they are of value. We included circ0084043 in our melanoma section, as it had a unique mechanism of action, was an excellent example of an under expressed circRNA pathology, and we believe had the most significant contribution to the paper. The recommended review written by De Palma et al (10.3390/biomedicines10030725) was utilized to expand our background text and was an excellent source of identification of primary sources, which were added and are also referenced. Similarly, the recommended review written by Xu et al. (doi: 10.3389/fgene.2022.1037120) was also of significant value, and was used to further explicate our background on gastric cancer as well as the addition of a very useful reference source for readers to explore.

CircRNAs described in the manuscript should be listed in a Table that also includes the tumor type, the sample description (i.e. tissue and/or plasma), the prognostic value, the expression status and the function (if known)

  • This table was made and included in our revision just prior to the conclusion. We agree this was of tremendous value for future readers.

There are some missing references. For example, Introduction, line 38 “In the last decade, a plethora of data have supported a significant role for ncRNAs in the development of many pathological conditions, especially cancer (REF)”, and line 60 “It has been described in various studies in the literature that circRNAs are actively contributing to carcinogenic propagation and progression

  • We agree these statements were lacking in references. Statements “In the last decade, a plethora of data have supported a significant role for ncRNAs in the development of many pathological conditions, especially cancer” was appropriately refrenced (Liu et al., 2021, Bhan et al., 2017, Bartonicek et al., 2016, Gupta et al., 2016) and “It has been described in various studies in the literature that circRNAs are actively contributing to carcinogenic propagation and progression was additionally referenced by the suggested source (doi: 10.1002/1878-0261.13034) and we agree including these references significantly increased the impact of the paper.

Thank you for your very helpful feedback, it was greatly appreciated. 

Best,

Nicole and Jan

Reviewer 2 Report

Comments and Suggestions for Authors

Comments and observations for the authors.

1)      The work is well written, but numerous literature and information is missing. For instance, despite the is a section focusing on AML, additional haematological tumors, such as multiple myeloma is missing (for instance authors can check https://cancerci.biomedcentral.com/articles/10.1186/s12935-023-03028-z). Same consideration can be made for solid tumors, although melanoma and breast cancer are described, additional tumor types such as gynaecological malignancies (https://www.frontiersin.org/articles/10.3389/fphar.2023.1194719/full), head and neck cancers (https://www.frontiersin.org/journals/oncology/articles/10.3389/fonc.2022.782439/full) in spite the fact that there is a vast literature in the field. Importantly, circRNAs also play a role in osteogenic differentiation (https://thno.org/v14p0143.htm) and osteosarcoma (https://www.ncbi.nlm.nih.gov/pmc/articles/PMC9850833/). For completeness, this information and references should be introduced in the text. Cumulatively, those are critical points that should be addressed

2)      Sub-head titles should be improved

3)      Important references in the field of circular RNAs and human cancer are missing and should be introduced

4)      A table summarizing al known circRNAs identified as diagnostic and prognostic markers in cancer being described in the review would be helpful

5) figures are well designed and clear

6) the table summarizing the currently kwown databases for circRNA in silico analysis of ceRNA axis is relevant and helpful

Comments on the Quality of English Language

English Language is good

Author Response

The work is well written, but numerous literature and information is missing. For instance, despite the is a section focusing on AML, additional haematological tumors, such as multiple myeloma is missing (for instance authors can check https://cancerci.biomedcentral.com/articles/10.1186/s12935-023-03028-z). Same consideration can be made for solid tumors, although melanoma and breast cancer are described, additional tumor types such as gynaecological malignancies (https://www.frontiersin.org/articles/10.3389/fphar.2023.1194719/full), head and neck cancers (https://www.frontiersin.org/journals/oncology/articles/10.3389/fonc.2022.782439/full) in spite the fact that there is a vast literature in the field. Importantly, circRNAs also play a role in osteogenic differentiation (https://thno.org/v14p0143.htm) and osteosarcoma (https://www.ncbi.nlm.nih.gov/pmc/articles/PMC9850833/). For completeness, this information and references should be introduced in the text. Cumulatively, those are critical points that should be addressed

  • We included a section on multiple myeloma as an additional source of haematological tumors, as we agree the paper extensively only covered solid tumors. Additionally, we greatly appreciated the suggestion to include gynaecological malignancies, which included a general introduction, specific introductions for endormetrial cancer, ovarian cancer, and cervical cancer. There is an abundance of circRNA research on ovarian cancer, and we were elated to have the suggestion to include it. Further, not much was identified for cervical cancer, which we feel is also important to highlight the lack of, and need for, research in that area. All authors are in full agreement of the contributed value of including these sections and references in the paper. Thank you.

Sub-head titles should be improved

  • Addressed for specifics of section coverage

Important references in the field of circular RNAs and human cancer are missing and should be introduced

  • We are in full agreement of this suggestion and to this, roughly 75 additional references were included in this revision text. We agree that this revised copy contains significantly more valuable sources for readers.

A table summarizing al known circRNAs identified as diagnostic and prognostic markers in cancer being described in the review would be helpful

  • Completed and included in revision

Figures are well designed and clear

  • Thank you.

The table summarizing the currently kwown databases for circRNA in silico analysis of ceRNA axis is relevant and helpful

  • Thank you- we agree this was a nice way to guide readers to the most used and valuable in silico sources used and referenced.

We are grateful for your helpful feedback. Thank you!

Best,

Nicole & Jan

Reviewer 3 Report

Comments and Suggestions for Authors

DeSouza et al. summrazied studies on circRNAs as potential cancer biomarkers. The topic is on interest, however, study suffers from some limitations:

1. As mentioned in the title, paper should focus on circRNAs as biomarkers, but authors described mainly mechanism of their action in various cancers. There are studies that directly measure role of this group RNAs as biomarkers in cancer tissues or/in blood samples. Also, diagnostic, predictive and prognostic values are available, as well. Authors should overthink structure ot their paper. Perphaps, could be better to divide its structure into diagnostic, prognostic and predictive value or something like that.

2. I don't fully agree with the figure 1 data. To be more clear - even under the physiological conditions some miRNAs are keept in low expression level to allow roboust trnaslation of some mRNAs.

3. To be more clear and readable, circRNAs should be summarized in table or tables with information of cancer type, group of patiets, etc.

4. Some circRNAs are described, why exactly those? The were seelcted from databases or found in literature?

5. Overall paper could sound more clinical, and some pros and cons of circRNAs in clinics should be discussed.

Author Response

As mentioned in the title, paper should focus on circRNAs as biomarkers, but authors described mainly mechanism of their action in various cancers. There are studies that directly measure role of this group RNAs as biomarkers in cancer tissues or/in blood samples. Also, diagnostic, predictive and prognostic values are available, as well. Authors should overthink structure ot their paper. Perphaps, could be better to divide its structure into diagnostic, prognostic and predictive value or something like that.

  • We agree the structure of our paper reads as a focus on molecular mechanisms of circRNAs in cancer. We adjusted the title to match our body of text, and we included statements, such as, Further, recent research has focused on circRNA targeting as a putative therapeutic approach (refs)……Genome-wide studies of a multitude of cancer types have revealed active pathologic roles for circRNA molecule expression and genomic modulation. It has been described in various studies in the literature that circRNAs are actively contributing to carcinogenic propagation and progression (refs)… ROC analysis recommendations are made in the literature to aid in determination of a circRNA’s potential as a putative biomarker” in order to explicate the notion that this research is imperative for the identification and use of circRNAs as emerging biomarkers.

I don't fully agree with the figure 1 data. To be more clear - even under the physiological conditions some miRNAs are keept in low expression level to allow roboust trnaslation of some mRNAs.

  • We agree and removed the title and hope our figure now exemplifies the basic roles of a ceRNA axis of regulation.

To be more clear and readable, circRNAs should be summarized in table or tables with information of cancer type, group of patiets, etc.

  • We are in agreement that this would be a tremendously helpful resource for readers and therefore included this table in our revision.

Some circRNAs are described, why exactly those? The were seelcted from databases or found in literature?

  • We included those that were recent, those that contributed to variation in overexpression and underexpression, those that have been found to have clinical value, those that had identified unknown molecular mechanisms and cellular interactions (i.e. with microRNAs), and those that were studied in, what we feel, a variety of tumor types and cancers of varying treatment/prognosis.

Overall paper could sound more clinical, and some pros and cons of circRNAs in clinics should be discussed.

  • We believe our revision included a more clinically relevant verbiage.

We kindly thank you for your great feedback and revisions. Wishing you all the best. 

Thank you,

Nicole and Jan

Round 2

Reviewer 1 Report

Comments and Suggestions for Authors

The authors have satisfactorily addressed my comments.

Reviewer 3 Report

Comments and Suggestions for Authors

In my opinion, the paper benefits from the revision and its quality is much better than before. I have no additional comments.